# *Streptococcus thermophilus*: To Survive, or Not to Survive the Gastrointestinal Tract, That Is the Question!

**DOI:** 10.3390/nu12082175

**Published:** 2020-07-22

**Authors:** Anđela Martinović, Riccardo Cocuzzi, Stefania Arioli, Diego Mora

**Affiliations:** Department of Food Environmental and Nutritional Sciences (DeFENS), University of Milan, via Celoria 2, 20133 Milan, Italy; andjela.martinovic@unimi.it (A.M.); riccardo.cocuzzi@unimi.it (R.C.); stefania.arioli@unimi.it (S.A.)

**Keywords:** *S. salivarius*, probiotics, taxonomy, recovery in fecal samples, microbiota

## Abstract

The probiotic market is increasing world-wide as well as the number of products marketed as probiotics. Among the latter, many products contain *Streptococcus thermophilus* strains at several dosages. However, the scientific evidence that should support the probiotic status of those *S. thermophilus* strains is often contradictory. This review analyses the scientific literature aimed to assess the ability of *S. thermophilus* strains to survive the human gastrointestinal tract by discussing the scientific validity of the methods applied for the bacterial recovery and identification from stool samples. This review highlights that in most of the intervention studies reviewed, the identification of *S. thermophilus* strains from stools was not carried out with the necessary taxonomic accuracy to avoid their misidentification with *Streptococcus salivarius*, a common human commensal and a species phylogenetically close to *S. thermophilus*. Moreover, this review highlights how critical the accurate taxonomic identification of *S. thermophilus* in metagenomics-based studies can be.

## 1. Introduction

According to the current definition of probiotics, “live microorganisms that, when administered in adequate amounts, confer a health benefit on the host” [1], the health-related probiotic traits (e.g., neurological, immunological, endocrinological effects) must be associated to a specific strain and depend on its viability *(Live microorganisms*) and dose administered (*Adequate amounts*). Nevertheless, there are also general, so-called *Core benefits* shared across a genus (e.g., regulation of intestinal transit, competitive exclusion of pathogens, normalization of perturbed microbiota) and species (e.g., direct antagonism, vitamin synthesis, enzymatic activity) level [1].

Most of the scientific literature dealing with the probiotic role of *Streptococcus thermophilus* strains starts from the assumption of the health benefit of yogurt consumption [2,3,4,5]. Yogurt is defined as a fermented milk based on the association of *S. thermophilus* and *Lactobacillus delbrueckii* subsp. *bulgaricus*. For this reason, strains belonging to these species have often been involved in scientific studies aimed to demonstrate their probiotic traits [6,7,8,9,10]. Due to its safe use in food production over the years, *S. thermophilus* was granted ‘Generally Recognized as Safe’ (GRAS) status in the USA and the ‘Qualified Presumption of Safety’ (QPS) status in the European Union [11]. Apart from being a constituent of yogurt, which has officially recognized probiotic properties (prerequisite is to contain at least 10^8^ colony-forming units (CFU)/g live microorganisms) [12], health benefits associated with *S. thermophilus* include production of antioxidant compounds, risk alleviation for some types of cancer, anti-inflammatory effects, antimutagenic effects and stimulation of the gut immune system [11,12,13,14]. The mitigating effect of yogurt consumption on lactose maldigestion has been officially recognized by European Food Safety Authority (EFSA) as a health claim [12]. In compliance with the Food and Agriculture Organization/World Health Organization (FAO/WHO) [15] modified definition of probiotics proposed by Hill and colleagues [1], such a beneficial effect has been attributed to the consumption of live bacterial cultures (*S. thermophilus* and *L. delbrueckii* subsp. *bulgaricus*) by several investigators [16,17,18]. For these reasons, probiotic products that include *S. thermophilus* strains are produced by multitudinous manufactures all over the world, such as those shown in Table 1. Most of the reported products (Table 1) are blends of several strains belonging to different species of lactic acid bacteria (e.g., *Lactobacillus rhamnosus*, *L. acidophilus*, *L. bulgaricus*, *L. reuteri*, *Bifidobacterium bifidum*, *B. breve*), *Bacillus* spp. (e.g., *Bacillus coagulans*, *Bacillus subtilis*, *Bacillus clauslii*) or the probiotic yeast *Saccharomyces boulardii*, with a CFU/dose ranging from 2 to 450 billion. Even though the market offers plenty of probiotic products containing *S. thermophilus* biomasses, the applicability of the term *probiotic* for this species itself is still questioned [19] due to its sensitivity to gastrointestinal tract (GIT) transit conditions.

The capability of surviving the GIT is generally considered a key feature of probiotic strains [1,20,21] and must be rigorously assessed. As highlighted by the FAO/WHO [15], typical in vitro tests such as resistance to gastric acidity and bile acid resistance require in vivo validation. One widespread way of doing so is the recovery of viable cells in fecal samples after oral administration, but are *S. thermophilus* strains easily recovered and correctly identified from stool samples? It is likely that other *Streptococcus* species, which inhabit the GIT, could be misidentified as *S. thermophilus*. Therefore, those taxonomic and ecological aspects should be considered in the intervention studies where *S. thermophilus* biomasses are administered, and in all the studies focused on the characterization of the human microbiota by metagenomics approaches. We review the scientific literature aimed to assess the ability of *S. thermophilus* strains to survive the human gastrointestinal tract by discussing the scientific validity of the methods applied for the bacterial recovery and identification from stool samples.

## 2. Existing Knowledge on *S. thermophilus* Survival in the Gastrointestinal Tract (GIT) Transit Conditions

Several studies attempted to assess the survival to the GIT of *S. thermophilus* by following different approaches, including culture-dependent and -independent methods or a combination of both [9,20,22,23,24,25,26,27,28] Mater et al. [20] quantified an orally administered streptomycin- and rifampicin-resistant *S. thermophilus* strain in fecal samples by standard plating, showing its substantial GIT survival in adults consuming yogurt. Similarly, Venturi et al. [22] described the recovery of *S. thermophilus* after oral administration of a multi-strain probiotic preparation by standard plating and also demonstrated the ability of *S. thermophilus* to survive the GIT. Subsequently, Brigidi et al. [23,24] and Mimura et al. [26] performed an identification and enumeration of *S. thermophilus* (combining both culture-dependent and -independent methods) from fecal samples after the administration of the same multi-strain probiotic product or yogurt, where they were able to recover significant numbers of *S*. *thermophilus* [23,24,26] as well as to demonstrate its colonization pattern [24]. By contrast, a study by del Campo and colleagues [25] dealing with the recovery of *S. thermophilus* from fecal samples using plating and polymerase chain reaction (PCR) assay provided negative results for the GIT survival of *S. thermophilus* in humans during a crossover study after daily yogurt consumption for two weeks. Afterwards, Elli et al. [27] again confirmed that *S. thermophilus* can be retrieved from human feces after yogurt consumption. Later on, García Hernández and colleagues [9] revealed the GIT survival and colonization pattern of *S. thermophilus* in some subjects by applying a direct viable count-fluorescence in situ hybridization (DVC-FISH) method to fecal samples.

Since those studies gave contradictory results on the ability of *S. thermophilus* strains to survive GIT conditions, some authors are still arguing if it should be considered as a probiotic, thus naming *S. thermophilus* a *Transient probiotic* [11]. It must be noted that those differences in GIT survival rates observed in different studies could have also been generated by the genetic and phenotypic diversity of *S. thermophilus* strains and/or by differences in fermentation process parameters adopted for the production of the biomasses or by differences in the absolute number of live cells administered [29].

## 3. Other Streptococci That Inhabit the Human GIT Could Interfere with the Correct Identification of *S. thermophilus* in Stool Samples

When attempting to recover *Streptococcus thermophilus* from fecal samples, one often neglected but critical aspect is its phylogenetic similarity with two other closely related species, namely *S. salivarius* and *S. vestibularis* [30]. These three species, which are genetically very similar [30], belong to the *S. salivarius* subgroup of viridians streptococci. *S. salivarius* and *S. vestibularis* are commensal bacteria of the oral and gastrointestinal cavities and of the genital tract [31]. The genetic similarity between *S. salivarius* and *S. thermophilus* was so high that the latter was previously classified as a subspecies of *S. salivarius* (*S. salivarius* subsp. *thermophilus*), and it obtained the status of species on its own only in 1991 [32]. *S. thermophilus* was described as an example of regressive evolution for the adaptation to a food environment [33]. *S. thermophilus* has lost all pathogenic traits throughout its evolution and adaptation to a well-defined, narrow ecological niche, the milk [33,34]. The regressive evolution of *S. thermophilus* was supported by the finding of a vestige of pathogenesis-related genes (10% of the total sequence), counterbalanced by the acquisition of relevant traits, like lactose and urea utilization, that have allowed the assembly of a new genome organization suitable for the colonization of the dairy niche [33,34]. The milk-driven speciation of *S. thermophilus* is also supported by the fact that this species is rarely, if ever, isolated outside the dairy environment [34]. However, the ability of *S. thermophilus* strains to utilize sucrose [34] with an efficiency comparable to that of lactose utilization led it to be hypothesized that this species could still be able to survive and grow in plant-based environments [35].

Sharing the same and being adapted to more natural and competitive niches, *S. salivarius* and *S. vestibularis* show a higher degree of physiological resemblance compared to *S. thermophilus.* For instance, *S. salivarius* and *S. vestibularis* have a higher capability to synthesize extracellular compounds (e.g., adhesins, exopolysaccharides) that are associated with bacterial fitness, resistance, biofilm formation and play a crucial role in cell adhesion, host colonization, or escape from the host immune system [30]. Moreover, the production of active urease, generally related to both *health-associated* and *virulence factors* is a common trait for all members of the *S. salivarius* group of streptococci [36]. The urease gene clusters of *S. thermophilus* and *S. salivarius* have similar organization (*ureIABCDEFGMQO*), with more complex transcriptional regulation in *S. salivarius* [37,38]. The well-known probiotic strain *S. salivarius* K12 is urease positive and can colonize the oral mucosae of humans regulating the innate immune responses of human epithelial cells [39]. In the case of the urease activity of *S. thermophilus*, Arioli and co-authors [40] observed that ammonia provided by urea hydrolysis boosted lactic acid production in both *S. thermophilus* and in *L. delbrueckii*, thus proposing urease activity as an altruistic cooperative trait. Regarding carbon sources utilization, *S. salivarius* can make use of a wider spectrum of compounds including glucose, fructose, galactose, lactose, as well as hydrolyze complex polymers such as starch or glucans [34]. On the other hand, *S. thermophilus* is well adapted to lactose as the main carbon source, hydrolyzing it to the glucose and galactose moieties. *S. thermophilus* is not able to metabolize galactose, expelling it outside the cell [34]. However, some galactose positive strains are reported [41]. Recently, multilocus sequence typing (MLST) and comparative genomic analysis gave novel insights into phylogeny, clearly separating these three species as well as indicating that *S. thermophilus* probably deviated from *S. vestibularis* [30,42].

Concerning the safety aspect, there is an overall consensus on the safety status of *S. thermophilus* species, whereas few reports highlighted that *S. salivarius* and *S. vestibularis* strains may cause mild to severe human infections [43,44].

*Streptococcus* spp. represent a notable part of the intestinal microbiota, with *S. salivarius* as one of the main species, indicating that these bacteria are actually able to colonize this ecological niche, rather than just being washed down with the flow of saliva [30,45,46,47]. *S. salivarius* has been recovered from fecal samples of infants, thus revealing that this species is one of the first inhabitants of the human gut [48,49]. Moreover, Delorme et al. [30] pointed out that previous metagenomic analysis of human feces revealed the presence of *S. salivarius* (formerly described as *S. thermophilus* due to a lack of appropriate genomic references) in more than 90% of the individuals tested [30,50].

In the light of these considerations, the answer to the question “Could *S. thermophilus* be misidentified with other streptococci that inhabit the human GIT?” is: yes, if the genomic similarities between these species are not considered (Figure 1).

## 4. The Methods Applied in the Reviewed Literature to Identify *S. thermophilus* Strains Were Not Selective at Species Level

We previously showed that some studies, dealing with the survival of *S. thermophilus* to the human GIT, based their assumptions on the recovery of *S. thermophilus* from the stool of human subjects after the consumption of yogurt or probiotic preparations [9,20,23,24,25,26,27,28]. Most of the studies indicated substantial [9,20,23,24,26,27,28] or no survival [25] of *S. thermophilus* strains after consumption of yogurt or probiotic products containing this species. What are the reasons for these contradictory conclusions? In most of the studies, the authors have paid little attention to the culture-dependent methods applied to recover live *S. thermophilus* strains from stool samples. The culture-dependent plating technique on specific selective media (e.g., ST agar, ST agar supplemented with the pH indicators bromocresole purple and bromocresole green, and the antibiotic nalidixic acid) used in some studies [22,23,24,28] was shown [23] to be ineffective for the selective identification and enumeration of *S. thermophilus* after the administration of a dairy or probiotic product, due to the inability to discriminate *S. thermophilus* from intestinal streptococci and enterococci as well as the phylogenetically closely related *S. salivarius* [23]. Therefore, the possibility to misidentify *S. thermophilus* with *S. salivarius*, the closest neighbor inhabiting the oral cavity, was likely underestimated. In addition, competitive inhibition by autochthonous bacteria from the gut, which may hinder the growth of *S. thermophilus* in the plate, cannot be excluded. Mater and colleagues [20] tried to selectively isolate *S. thermophilus* S85, which was streptomycin and rifampicin resistant, by adding the two antibiotics to the medium used for the recovery from fecal samples; however, the sensitivity of *S. salivarius* strains to streptomycin and rifampicin was not verified, nor was a molecular identification performed on the isolated colonies to confirm their taxonomy. Therefore, to date, the data about the accurate identification and retrieval of *S. thermophilus* in human subjects upon administration of yogurt or probiotic preparations are debatable.

### 4.1. Are the Data Supporting the GIT Survival of S. thermophilus Strains Solid and Unbiased Enough to Generate Scientific Consensus?

A list of molecular tools developed for the identification of *S. thermophilus*, together with the limitations detected for each of them, is shown in Table 2. Some of the tools listed in Table 2 have been developed aiming to identify *S. thermophilus* from fermented dairy products. These tools are based on species-specific polymerase chain reaction (PCR) primers targeting the *16S rDNA* gene sequence [51,52]. However, the *16S rDNA* gene sequence is not specific enough because of a high identity among those sequences in *S. thermophilus*, *S. salivarius* and *S. vestibuilaris* species (99.8% between *S. thermophilus* and *S. salivarius*; 99.6% between *S. thermophilus* and *S. vestibularis*) [53]. Lick and Teuber [54] designed species-specific oligonucleotide probes based on the 5′ region of the *lacZ* gene, but they showed both specific and non-specific hybridization signals among different strains of *S. thermophilus* and *S. salivarius*. Later on, species-specific PCR primers targeting the *lacZ* gene were developed [55]. This primer set was validated using several *S. thermophilus* strains and two *S. salivarius* strains as a reference [55]. However, subsequent studies demonstrated the inability of *lacZ* gene primers to correctly identify *S. thermophilus* strains [27].

In 1997, Tilsala-Timisjärvi and Alatossava [56] designed the ThI/ThII PCR species-specific primer set, based on the internal transcribed spacer between the *16S* and the *23S rDNA* genes for the detection of *S. thermophilus* in dairy products. In this study, the authors underlined that this primer set was most probably not able to distinguish between *S. thermophilus* and *S. salivarius*, which, being a closely related species, has an almost identical spacer sequence between the *16S rDNA* and the *23S rDNA* genes. Subsequently, Mora and colleagues [41], analyzing the sequence of the *16S-23S rDNA* spacer region, demonstrated that a simple end-point PCR assay carried out using ThI/ThII primers was ineffective to discriminate between *S. salivarius*, *S. thermophilus*, *S. bovis*, *S. macedonicus*, *S. waius* and *S. infantarius* species. Nevertheless, the two primers were used in various intervention studies to carry out the identification of *S. thermophilus* strains recovered from stool samples [23,24,26]. More specifically, Brigidi et al. [23] performed an identification and enumeration of *S. thermophilus* from fecal samples of individuals who had been taking a multi-strain probiotic product using ThI/ThII primers, reporting a significant increase of *S. thermophilus* after administration of this probiotic product. Later, the same group of authors conducted a similar study using a culture-independent quantitative polymerase chain reaction (qPCR) method with the same primer set (ThI/ThII) for the identification and enumeration of several *S. thermophilus* strains showing its survival as well as colonization pattern after administration of both a probiotic preparation and commercial yogurt [24]. However, quantitative PCR of *S. thermophilus* was applied directly on feces, thus it was not possible to discriminate between live and dead cells as well as free DNA originated from lysed cells. In this case, a propidium monoazide (PMA-qPCR)-based methodology [57] would have been appropriate to directly quantify live *S. thermophilus* cells excluding dead cells and free DNA from the fecal samples. Moreover, in this study the primers where accurately tested for specificity against some intestinal and dairy species, among them *S. faecium* (several strains), *Lactobacillus acidophilus* (several strains), *L. brevis*, *L. casei*, *L. crispatus*, *L. delbrueckii* subsp. *delbrueckii* (several strains), *L. delbrueckii* subsp. *lactis*, *Lactococcus lactis*, (several strains), *L. lactis* subsp. *cremoris* (several strains), *L. lactis* subsp. *lactis* biovar *diacetylactis*, *Escherichia coli* (several strains), *Enterococcus fecalis* (several strains), *Clostridium beijerinkii*, *C. perfringens*, *Bacteroides fragilis* (several strains), *Bifidobacterium bifidum* (several strains), *B. infantis* (several strains), *B. longum* (several strains), *B. adolescentis* (several strains) but not against the phylogenetically close *S. salivarius*. Similarly, Mimura et al. [26] demonstrated higher levels of *S. thermophilus* after administration of a multi-strain probiotic product using the same primer set. These studies, based on the use of ThI/ThII primers, also showed that some of the subjects harbored *S. thermophilus* even after the run in and prior to treatment period, where the subjects were asked not to consume any food products containing this species. Therefore, we can speculate that primers may have actually detected *S. salivarius* strains, providing misleading results.

Contrary to previous data, del Campo and colleagues [25] provided scarce evidence of *S. thermophilus* recovery in human feces after daily yogurt consumption. In this study, not a single positive amplification for *S. thermophilus* was obtained from the total fecal DNA of 114 volunteers. The authors attempted to detect *S. thermophilus* using the THER-F/THER-R primer set, which is targeted to the *lacZ* gene. Nevertheless, due to some weaknesses in the protocols adopted for DNA extraction from fecal samples, the results obtained may have not been correctly analyzed. In addition, the specificity of the THER-F/THER-R primers for *S. thermophilus* was questioned in a later study by Elli and colleagues [27] who demonstrated that this primer set was not specific enough to set apart *S. salivarius* and detect exclusively *S. thermophilus*. Due to the lack of efficient identification methods, the same authors carried out a pulsed-field gel electrophoresis (PFGE) based on different SmaI digested chromosomes, in order to distinguish these two genetically close species and recover *S. thermophilus* from the feces of 20 healthy volunteers who had been consuming commercial yogurt. Likewise, it was suggested that there is a need for reliable molecular tools for the identification of orally administered *S. thermophilus* strains. In a 2012 study, García Hernández and colleagues [9] for the first time applied the DVC-FISH method to fecal samples for the elucidation of the survival rate of *S. thermophilus* strain CECT 801 to GIT transit. It must be observed, however, that the FISH probe used in the study, namely STH23 targeting the *23S rRNA* gene, may not be specific enough for *S. thermophilus* because only one single base mismatch was present between the FISH probe target region of *S. thermophilus* and *S. salivarius.* In 2009, Ongol et al. [58] developed a real-time PCR method targeting a gene encoding a 16S rRNA processing protein (*rimM*) for the detection of *S. thermophilus* in dairy products. The authors tested the primer set against several lactic acid bacteria, including *S. salivarius*, demonstrating its high specificity but failed to amplify some *S. thermophilus* strains.

In light of the scientific literature cited we can, therefore, conclude that to date there are no taxonomically correct assays that allow to unambiguously identify *S. thermophilus* strains and distinguish them from those belonging to *S. salivarius* in a stool sample.

### 4.2. Culture-Independent Detection and Identification of S. thermophilus

The problems related to the incorrect identification of *S. thermophilus* are not limited to the studies aimed to verify the recovery of live cells in stool samples during intervention studies. We are living in the microbiota era, and the role of microorganisms is becoming increasingly relevant in all ecosystems, including the human ecosystem. Several studies have been focused on the role of diet- and food-associated microorganisms on gut microbiota and human health. The complexity of the taxonomy of *S. thermophilus*/*S. salivarius*/*S. vestibularis* group was often completely ignored also in this research field. Francavilla and colleagues [59], in a study aimed to characterize the salivary microbiota and metabolome associated with celiac disease, reported that *S. thermophilus* levels in human saliva were markedly decreased in a group of subjects that undergo a gluten-free diet. In this study saliva microbiota was characterized by *16S rRNA* gene amplicon library sequencing, and the authors did not take into consideration both the taxonomy and the ecology of the environment they studied, making the readers erroneously believe that *S. thermophilus* is a common inhabitant of our oral cavity. More recently, Pasolli and colleagues [60] published an interesting study aimed to elucidate to what extent the lactic acid bacteria we ingest through the consumption of fermented food become members of the gut microbiome. The authors analyzed 9445 metagenomes from human samples and demonstrated that the prevalence and abundance of lactic acid bacteria species in stool samples is generally low and linked to age, lifestyle, and geography, with *S. thermophilus* being one of the most prevalent species [60]. To reach this conclusion, the authors focused on a defined list of genomes to be used as reference for the quantification of the prevalence of lactic acid bacteria species in the human gut, but unfortunately only the genome of *S. thermophilus* was present in the list, whereas the genome of *S. salivarius* was not considered at all [60] although the latter is recognized as a major human commensal [30]. Therefore, how can we provide adequate scientific evidence towards the statement that the increased consumption of yogurts and other dairy products are the sources of *S. thermophilus* in our gut ecosystem? Moreover, which metabolic features of *S. thermophilus* would guarantee adaptation of this species to the human gut environment? Why did the authors not consider that *S. thermophilus* was described as an example of genomic regressive evolution for adaptation to the milk environment [33,34]? In other studies [61,62] aimed to find correlation between the diet and the gut microbiota composition, where the proper reference genomes have been used, *S. thermophilus* was not considered as a human gut inhabitant.

## 5. Conclusions

Growing interest in the probiotic market, together with the increasing number of single and multi-strain products, is not always accompanied by scientific studies aimed to address the health benefit on the host of such products. In the last few years, some efforts have been made to evaluate the quality levels of products that are commercialized [63,64,65,66,67]. However, limited efforts have been made to develop molecular or phenotypic tools able to accurately identify the species and/or the strains administered as probiotics in intervention studies. The accurate taxonomic identification is, therefore, a paramount issue to be validated for the compliance of probiotic food and supplement products. Without a correct taxonomy assignment of the species/strains that should contain probiotic traits, the interpretation of the results obtained from the intervention studies is thwarted. This is especially the case of *S. thermophilus*, whose probiotic status has been in debate for decades. Being *S. thermophilus* a species with high economic relevance for the probiotic market, and *S. salivarius* one of the major human commensals able to colonize several human body-related sites, including the digestive tract [30], the correct identification of these taxa should be of high interest for the food and pharma industry. Another important issue arising from the reviewed literature is the absence of guidelines and protocols for the recovery from stool samples of orally ingested probiotics. The guidelines should recommend a proper study design based on a double-blind, placebo-controlled crossover design, an appropriate washout period together with the application of a species-specific plate culture method (if available), and/or the use of strain-specific molecular tools to correctly identify the orally administered probiotic/s. Moreover, among the commonly adopted exclusion criteria used in the intervention studies (e.g., antibiotic consumption prior to the start of the study, consumption of antacids or prokinetic gastrointestinal drugs, history or presence of significant intestinal diseases) the exclusion of the subjects showing a positive detection of probiotic strain in fecal samples during the run in or the placebo intake period should be considered because it might represent a possible compliance violation due to unauthorized probiotic intake.

## Figures and Tables

**Figure 1 nutrients-12-02175-f001:**
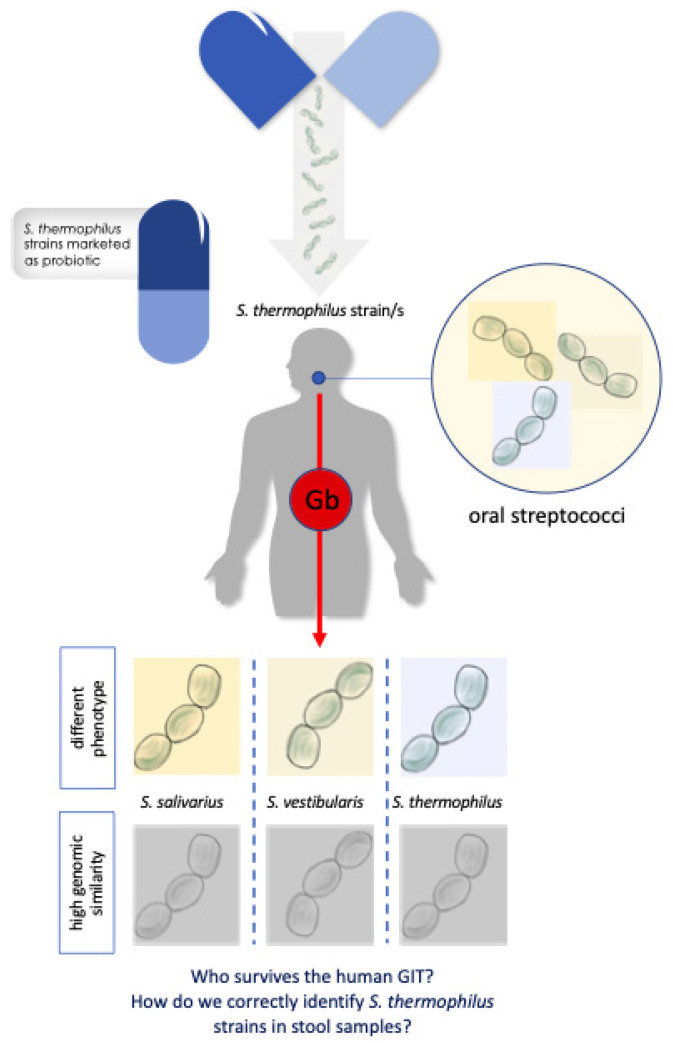
Schematic representation of the criticisms associated to the identification of *S. thermophilus* strains in human stool samples during intervention studies. Gb, gastric barrier; GIT, gastrointestinal tract.

**Table 1 nutrients-12-02175-t001:** List of chosen probiotic products available on the international market containing *S. thermophilus* strains.

Product Brand Name	*S. thermophiles* Strain	Manufacturer	Colony-Forming Units (CFU)/Dose	Country of Origin
Fermental	n.d.	ESI S.r.l.	2 bilions/capsule	Italy
Floratrex	n.d.	Global Healing Center	75 bilion/0.8 g	USA
Lactoflor AB	n.d.	Global Medic	500 billion/capsule	Ecuador
Lactoflor Probiotic	n.d.	Lactoflor	2 billion/0.4 g	Bulgaria
Multibiotics	n.d.	NutriZing	50 billions/g (capsules)	UK
Neuflor	Z57 (BCCM: LMG P-21908)	Yamamoto nutrition	56 billion/0.8 g	Italy
Nexabiotic Advanced	n.d.	DrFormulas	17.25 billion/capsule	USA
NOW Probiotic-10	St-21	NOW Foods	50 billion/capsule	USA
Oti Probioti	SP4	O.T.I. Officine Terapie Innovative S.r.l.	25.2 billion/4 capsules	Italy
Perfect Biotics	n.d.	Probiotic America	30 billion/capsule	USA
Pro-100 Ultra	n.d.	Natures Aid	100 billion/capsule	UK
PRO-30 Max	n.d.	Natures Aid	30 billion/capsule	UK
PRO-B ACTIVE	n.d.	Lilly Drogerie	14 bilions/capsule	Serbia
PROBIO7	n.d.	PROBIO	4 billion/capsule	UK
Probioguard	n.d.	Lamberts	4 billion/capsule	UK
Probiona Plus	n.d.	Tauron Ventures GmbH	30 billion/capsule	Germany
Probiotic supplement with vitamins	SD5207	L’Angelica Istituto Erboristico	18 billions/1.8 g (soluble sticks)	Italy
Prodefen	PXN 66	Italfarmaco S. A	10 billion *L. rhamnosus* +1 bilion (other strains)/1 g (sachets)	Spain
Protexin Bio-Kult	PXN 66	Probiotics International L.t.d.	10 bilion/g (capsule)	UK
Synbiotic365	UASt-09	United Naturals	20 billion/capsule	Canada
Ultra-potency	ST-21	Custom Probiotics	320 billion/0.8 g (powder)	USA
Visbiome	DSM24731	ExeGi Pharma	1125 billion/g (capsule)	USA
Vivomixx	DSM 24731	Mendes S.A.	450 billion/4.4 g (sachet)	Switzerland
VSL#3	BT01	Actial Farmaceutica S.r.l.	450 billion/4.4 g (sachets) 112 billion/658 mg (capsules) 45 billion/1.5 g (sticks)	Italy
Yovis	Z57 (BCCM: LMG P-219089)	AlfaSigma	50 billion/1.5 g (sachets) 50 billion/0.7 g (capsules) 25 billion/10 mL (falcons)	Italy

Abbreviations: n.d.: strain code not declared; DSM: German Collection of Microorganisms and Cell Cultures GmbH; BCCM: Belgian Co-ordinated Collections of Micro-organisms.

**Table 2 nutrients-12-02175-t002:** Overview of tools used in interventional studies for viability assessment of *S. thermophilus.*

Product	Used Tool	Target Species/Gene	Species-Specificity	Other Potentially Detected Species	Reference
Probiotic food supplement	Standard plating	*S. thermophilus*	no	*S. salivarius*, *S. vestibularis*	[22]
Probiotic food supplement	End-point PCR	*16S-23S rDNA* spacer region	no	*S. salivarius*, *S. vestibularis*	[23]
Probiotic food supplement and Yogurt	End-point PCR, qPCR	*16S-23S rDNA* spacer region	no	*S. salivarius*, *S. vestibularis*	[24]
Probiotic food supplement	End-point PCR	*16S-23S rDNA* spacer region	no	*S. salivarius*, *S. vestibularis*	[26]
Yogurt	End-point PCR, DNA-hybridization	*lacZ* gene	no	*S. salivarius*, *S. vestibularis*	[25]
Yogurt	End-point PCR, PFGE	*lacZ gene*, Profiles of SmaI-digested chromosomes	no	*S. salivarius*, *S. vestibularis*	[27]
Yogurt	DVC-FISH	*23S rRNA*	no	*S. salivarius*, *S. vestibularis*	[9]

Abbreviations: PCR: polymerase chain reaction; qPCR: quantitative polymerase chain reaction; PFGE: pulsed-field gel electrophoresis.

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
