# Peer review of "Streptococcus thermophilus: To Survive, or Not to Survive the Gastrointestinal Tract, That Is the Question!"

_nutrients, 2020, doi:10.3390/nu12082175_

Round 1

Reviewer 1 Report

This review demonstrates that several studies showing that Streptococcus. thermophilus can reach the human gastrointestinal tract alive are misinterpreted, because there is a misidentification between S. thermophilus and S. salivarious, an intestinal bacteria, due to no suitable PCR primer set that can be correctly discriminate between both species in feces.

Major comments

1. This manuscript should be checked by an English speaker, as there is a few grammatical errors.

2. In Line 83, "Materials and Methods" is incorrect. It should be revised appropriately. For example, it is recommended to revise "Materials and Methods" to "Results and Discussion" and to show Table 1 in "Materials and Methods".

3. In Line 221-222, a culture independent qPCR method should not be used for fecal recovery studies of ingested probiotics, because it detects not only viable cells but also cells that have DNA but cannot form colonies, although there is a problem with the strain specificity of the PCR primer set. A propidium monoazide-qPCR method, which can detect viable cells only, should be used to solve the problem.

4. This is not just an issue of S. thermophiles. It is considered that the most important issue is the lack of a guideline that demonstrates a valid test protocol for fecal recovery of orally ingested probiotics. The test protocol should include a double-blind, placebo-controlled crossover design, an appropriate washout period, a species-specific plate culture method, and a strain-specific PCR primer set. If a probiotics strain is detected in feces of some subjects during the placebo intake period, these subjects should be excluded because of an insufficient washout period or possible compliance violations such as probiotic intake. If a probiotic strain is detected in feces of many subjects during the placebo intake period, a specificity of the primer set may be suspected, and the test is considered to be unsuccessful. This issue should be described in the manuscript.

Author Response

Review #1

Comments and Suggestions for Authors

This review demonstrates that several studies showing that Streptococcus. thermophilus can reach the human gastrointestinal tract alive are misinterpreted, because there is a misidentification between S. thermophilus and S. salivarious, an intestinal bacteria, due to no suitable PCR primer set that can be correctly discriminate between both species in feces.

Major comments

1.This manuscript should be checked by an English speaker, as there is a few grammatical errors.

Author response: Grammar errors have been checked and corrected;

  1. In Line 83, "Materials and Methods" is incorrect. It should be revised appropriately. For example, it is recommended to revise "Materials and Methods" to "Results and Discussion" and to show Table 1 in "Materials and Methods".

Author response: Materials and methods title was a mistake of manuscript editing. The manuscript file we submitted was not formatted as the one you reviewed. Anyway, the revised version of the manuscript was corrected accordingly;

  1. In Line 221-222, a culture independent qPCR method should not be used for fecal recovery studies of ingested probiotics, because it detects not only viable cells but also cells that have DNA but cannot form colonies, although there is a problem with the strain specificity of the PCR primer set. A propidium monoazide-qPCR method, which can detect viable cells only, should be used to solve the problem.

Author response: The qPCR methods mentioned in the reviewed literature was applied to cultivated biomasses. We agree that qPCR cannot detect viable microorganism as it is possible for PMA-based PCR applications;

  1. This is not just an issue of S. thermophilus. It is considered that the most important issue is the lack of a guideline that demonstrates a valid test protocol for fecal recovery of orally ingested probiotics. The test protocol should include a double-blind, placebo-controlled crossover design, an appropriate washout period, a species-specific plate culture method, and a strain-specific PCR primer set. If a probiotics strain is detected in feces of some subjects during the placebo intake period, these subjects should be excluded because of an insufficient washout period or possible compliance violations such as probiotic intake. If a probiotic strain is detected in feces of many subjects during the placebo intake period, a specificity of the primer set may be suspected, and the test is considered to be unsuccessful. This issue should be described in the manuscript.

Author response: we agree with the reviewer observation, but the lack of guidelines for a valid test protocol for fecal recovery is another issue that is not considered as a focus of our review.

Reviewer 2 Report

The manuscript entitled “Streptococcus thermophilus: to survive, or not to survive the gastrointestinal tract, that is the question!” submitted for revision in Nutrients had been positively reviewed with some minor modifications.

This is an interesting manuscript describing the probiotic role of S. thermophilus strains and assessing the ability of S. thermophilus strains to survive in the human digestive tract. The authors also note the effectiveness of the tools used to identify species in stool samples.

There are little research and information on the molecular or phenotypic tools capable of accurate taxonomic identification of the probiotic strain (s) during intervention tests. The Authors suggest that the correct identification of these taxa should be of great interest in the food and pharmaceutical industries because S. thermophilus is a species of high economic importance for the probiotic market.

The authors in a paper try to answer the questions

Are S. thermophilus strains able to survive the human gastrointestinal tract?

Could S. thermophilus be misidentified with other streptococci that inhabit the human GIT?

Was S. thermophilus correctly identified from stool samples?

Is S. thermophilus a common inhabitant of the human GIT?

The review is positive. Therefore, I propose minor amendments.

  1. Keywords should be different from the publication title. Delete: Streptococcus thermophiles, gastrointestinal tract
  2. In title Table 1. I suggest adding “chosen” because there are many products (supplements) in markets. Table 1. List of chosen probiotic products available on the international market containing S. thermophilus strains.

In the column: S. thermophiles strain please correct the notation n.d., sometimes the font is bold, sometimes italic.

I think that the Single / Multi-strain column is not needed because Single occurs only once and it is enough not to enter anything in the column: Other strains

  1. No title 2. Materials and methods needed because this is a review publication
  2. There is a lack of literature from the last 3 years. Authors may also include the latest publication ex.:

Cécile Philippe, Sébastien Levesque, Moïra B. Dion, Denise M. Tremblay, Philippe Horvath, Natascha Lüth, Christian Cambillau, Charles Franz, Horst Neve, Christophe Fremaux, Knut J. Heller, Sylvain Moineau. Novel Genus of Phages Infecting Streptococcus thermophilus: Genomic and Morphological Characterization. Applied and Environmental Microbiology Jun 2020, 86 (13) e00227-20; DOI: 10.1128/AEM.00227-20

Hu, T., Cui, Y., Zhang, Y., Qu, X., & Zhao, C. (2020). Genome Analysis and Physiological Characterization of Four Streptococcus thermophilus Strains Isolated From Chinese Traditional Fermented Milk. Frontiers in microbiology, 11, 184. https://doi.org/10.3389/fmicb.2020.00184

Author Response

Review #2

Comments and Suggestions for Authors

The manuscript entitled “Streptococcus thermophilus: to survive, or not to survive the gastrointestinal tract, that is the question!” submitted for revision in Nutrients had been positively reviewed with some minor modifications.

This is an interesting manuscript describing the probiotic role of S. thermophilus strains and assessing the ability of S. thermophilus strains to survive in the human digestive tract. The authors also note the effectiveness of the tools used to identify species in stool samples.

There are little research and information on the molecular or phenotypic tools capable of accurate taxonomic identification of the probiotic strain (s) during intervention tests. The Authors suggest that the correct identification of these taxa should be of great interest in the food and pharmaceutical industries because S. thermophilus is a species of high economic importance for the probiotic market.

The authors in a paper try to answer the questions

Are S. thermophilus strains able to survive the human gastrointestinal tract?

Could S. thermophilus be misidentified with other streptococci that inhabit the human GIT?

Was S. thermophilus correctly identified from stool samples?

Is S. thermophilus a common inhabitant of the human GIT?

The review is positive. Therefore, I propose minor amendments.

1-Keywords should be different from the publication title. Delete: Streptococcus thermophiles, gastrointestinal tract

Author response: We changed the key words accordingly to the reviewer indications (line 20).

2-In title Table 1. I suggest adding “chosen” because there are many products (supplements) in markets. Table 1. List of chosen probiotic products available on the international market containing S. thermophilus strains.

Author response: We changed the title of Table 1 as suggested by the reviewer (line 80);

In the column: S. thermophilus strain please correct the notation n.d., sometimes the font is bold, sometimes italic.

Author response: Corrections has been made as suggested by the reviewer;

I think that the Single / Multi-strain column is not needed because single occurs only once and it is enough not to enter anything in the column: Other strains

Author response: We agree with the reviewer suggestions. Therefore, the columns “single/multi-strain” and “other strains” have been deleted from Table 1.

3-No title 2. Materials and methods needed because this is a review publication

Author response: Materials and methods title was a mistake of manuscript editing. The manuscript file we submitted was not formatted as the one you reviewed. Anyway, the revised version of the manuscript was corrected accordingly;

There is a lack of literature from the last 3 years. Authors may also include the latest publication ex.:

Cécile Philippe, Sébastien Levesque, Moïra B. Dion, Denise M. Tremblay, Philippe Horvath, Natascha Lüth, Christian Cambillau, Charles Franz, Horst Neve, Christophe Fremaux, Knut J. Heller, Sylvain Moineau. Novel Genus of Phages Infecting Streptococcus thermophilus: Genomic and Morphological Characterization. Applied and Environmental Microbiology Jun 2020, 86 (13) e00227-20; DOI: 10.1128/AEM.00227-20

Hu, T., Cui, Y., Zhang, Y., Qu, X., & Zhao, C. (2020). Genome Analysis and Physiological Characterization of Four Streptococcus thermophilus Strains Isolated From Chinese Traditional Fermented Milk. Frontiers in microbiology, 11, 184. https://doi.org/10.3389/fmicb.2020.00184

Author response: Despite the reviewer suggestion, the authors do not understand why the two suggested references should be cited in the review manuscript because they are not related to the probiotic applications of S. thermophilus or to the development of species-specific molecular tools. Therefore, the two references have not been added to the reference list of the manuscript.

Reviewer 3 Report

Please identify what is the most important issue that you are looking at with this review: survival in the GIT or phylogenetic identification or a combination of both. This must be clear throughout the text while right now it is not. Please review the manuscript format, e.g. the M&M here are confusing. What about Discussion? Please revise and use appropriate titles capturing the significant points of your review. A major taxonomic reclassification on probiotics was published recently (Zheng et al. 2020); maybe your discussion could benefit from this. 

Minor grammar/language corrections are needed e.g. in lines 37-40 the syntax needs corrections. Line 95: 'numbers of'. Lines 95-96: Please rephrase 'on the other side' into a more formal way; 'recovery' from?please specify. Line 137:'Concerning' must be rephrased. Line 144:'Mostly' does not read well, please rephrase. Line 200: 'because of'.

Please mention in your Abstract what will you review.

Lines 51-52: Please discuss more the data in Table 1 to better justify its presence in the manuscript.

Lines 55-59: This sentence has two parts not clearly connected with 'but'. One issue is the tests required. Another issue is how the organism is isolated. The two are not connected. Please rephrase.

Lines 53-63: Please make clear in what are the issues regarding the 'probiotic' state of the organism. The main is if it survives the GI track. How is this affected by the way it's being isolated, identified etc. Please rephrase in a clear way where you state the existing issues clearly and what will you assess in the following paragraphs maybe without the question marks but as clear statement.

Line 83: Is this title M&M related to how you conducted the review search? I think you refer to M&M of the studies you reviewed? Please check if so, rephrase the title bcs as it is now, it seems as if you refer to M&M that you followed. 

Lines 96-101: Please add more details such as how regular was the yogurt consumption, or specify 'in some of the subjects'.

Line 108: Please specify what GIT stands for in your manuscript; discrepancy with line 55.

Line 101:'similarity' may refer to morphology but I think you refer to phylogeny. Please specify this. Please write the name in full if it's the first time they appear in the text.

Line 112, 116-117, 125-126, 127-128, 134, 143-144, 205: reference please.

Line 153: Please specify what you mean by 'true development'; it is not clear.

Lines 174-187: Please clarify: So, the studies you refer to have isolated some strains in culture but 16S methods (or what methods?) could not identify which strain or species they were? Please specify and explain bcs it is not clear. You haven't referred to identification methods so far and this is important. You do it further down but not mentioning it so far creates vagueness. 

Lines 195-196: Please rephrase the '..we be convinced...' bcs it gives a non-scientific tone.

Line 173: Please break this long paragraph into smaller ones with clear content. Maybe re-title in combination to comment for line 83.

Line 278: Are these ... there for a reason?

Lines 271-272: Here you seem to report a key problematic of the studies. Should this be detailed further in the text and Intro?

Author Response

Review #3

Comments and Suggestions for Authors

Please identify what is the most important issue that you are looking at with this review: survival in the GIT or phylogenetic identification or a combination of both. This must be clear throughout the text while right now it is not.

Author response: The abstract has been modified accordingly to the reviewer’s suggestions highlighting the most important issue of the review (line 12-19).

Please review the manuscript format, e.g. the M&M here are confusing. What about Discussion? Please revise and use appropriate titles capturing the significant points of your review. A major taxonomic reclassification on probiotics was published recently (Zheng et al. 2020); maybe your discussion could benefit from this.

Author response: Materials and methods title was a mistake of manuscript editing. The manuscript file we submitted was not formatted as the one you reviewed. Anyway, the revised version of the manuscript was corrected accordingly; All titles have been re-written in order to capturing the significant points of the text. All paragraphs contain analysis of the data collected from the literature and an appropriate discussion. Therefore, we have decided to maintain only the “Conclusion” section.

Minor grammar/language corrections are needed e.g. in lines 37-40 the syntax needs corrections. Line 95: 'numbers of'. Lines 95-96: Please rephrase 'on the other side' into a more formal way; 'recovery' from?please specify. Line 137:'Concerning' must be rephrased. Line 144:'Mostly' does not read well, please rephrase. Line 200: 'because of'.

Author response: All minor grammar/language corrections have been done (line 12-19).

Please mention in your Abstract what will you review.

Author response: The abstract has been modified accordingly to the reviewer’s suggestions (lines 12-15), as well as introduction (line 61-64)

Lines 51-52: Please discuss more the data in Table 1 to better justify its presence in the manuscript.

Author response: The Table has been modified accordingly to the reviewer’#1 suggestion, and a short discussion has been added in the text as suggested by reviewer #2 (line 44-52).

Lines 55-59: This sentence has two parts not clearly connected with 'but'. One issue is the tests required. Another issue is how the organism is isolated. The two are not connected. Please rephrase.

Author response: The sentence was re-written (line 55-57).

Lines 53-63: Please make clear in what are the issues regarding the 'probiotic' state of the organism.

Author response: The sentence was re-written focus the attention on the human GIT survival (line 55-58);

The main is if it survives the GI track. How is this affected by the way it's being isolated, identified etc. Please rephrase in a clear way where you state the existing issues clearly and what will you assess in the following paragraphs maybe without the question marks but as clear statement.

Author response: The sentence was re-written accordingly with the reviewer’s comments and suggestions (line 55-64);.

Line 83: Is this title M&M related to how you conducted the review search? I think you refer to M&M of the studies you reviewed? Please check if so, rephrase the title bcs as it is now, it seems as if you refer to M&M that you followed.

Author response: Materials and methods title was a mistake of manuscript editing. The manuscript file we submitted was not formatted as the one you reviewed. Anyway, the revised version of the manuscript was corrected accordingly;

Lines 96-101: Please add more details such as how regular was the yogurt consumption, or specify 'in some of the subjects'.

Author response: More details have been added to the text (line 95-98).

Line 108: Please specify what GIT stands for in your manuscript; discrepancy with line 55.

Author response: GIT meaning has been specified to avoid discrepancy across the manuscript.

Line 101:'similarity' may refer to morphology but I think you refer to phylogeny. Please specify this.Please write the name in full if it's the first time they appear in the text.

Author response: We corrected the text accordingly to the reviewer’s comment (line 112).

Line 112, 116-117, 125-126, 127-128, 134, 143-144, 205: reference please.

Author response: All requested references have been added (line 111-128).

Line 153: Please specify what you mean by 'true development'; it is not clear.

Author response: The sentence was modified as requested by the reviewer (line 155).

Lines 174-187: Please clarify: So, the studies you refer to have isolated some strains in culture but 16S methods (or what methods?) could not identify which strain or species they were? Please specify and explain bcs it is not clear. You haven't referred to identification methods so far and this is important. You do it further down but not mentioning it so far creates vagueness.

Author response: We thank the reviewer for the comment. Here we refer to the culture-dependent methods as now underlined in the text of the revised version of the manuscript.

Lines 195-196: Please rephrase the '..we be convinced...' bcs it gives a non-scientific tone.

Author response: The sentence was re-written (line 200).

Line 173: Please break this long paragraph into smaller ones with clear content. Maybe re-title in combination to comment for line 83.

Author response: The paragraph was divided in two sections (section 4 and 4.1);

Line 278: Are these ... there for a reason?

Author response: The”…” have been deleted. It was a typo (line 128).

Lines 271-272: Here you seem to report a key problematic of the studies. Should this be detailed further in the text and Intro?

Author response: The relevance of correct taxonomic identification of S. thermophilus in metagenomic-based studies was mentioned in the new version of the abstract and the introduction sections (line 18-19; 60-61).

Round 2

Reviewer 1 Report

This review demonstrates that several studies showing that Streptococcus. thermophilus can reach the human gastrointestinal tract alive are misinterpreted, because there is a misidentification between S. thermophilus and S. salivarious, an intestinal bacteria, due to no suitable PCR primer set that can be correctly discriminate between both species in feces. It is considered to be socially significant to publish this review, because manufacturers of probiotics need to disclose the correct information to consumers. However, it need some additional discussion.

Major comments

  1. In Line 228-231, a culture independent qPCR method is described. This method should not be used for fecal recovery studies of ingested probiotics, because it detects not only viable cells but also cells that have DNA but cannot form colonies, although there is a problem with the strain specificity of the PCR primer set. A propidium monoazide-qPCR method, which can detect viable cells only, should be used to solve the problem. You should discuss this problem.
  2. Not only the problem of S. thermophilus, but the lack of guidelines that demonstrates an appropriate test protocol for the recovery of orally ingested probiotics from feces is considered to be the most important problem. The test protocol should include a double-blind, placebo-controlled crossover design, an appropriate washout period, a species-specific plate culture method, and a strain-specific PCR primer set. And if a probiotics strain is detected in feces of some subjects during the placebo intake period, these subjects should be excluded because of an insufficient washout period or possible compliance violations such as probiotic intake. If a probiotic strain is detected in feces of many subjects during the placebo intake period, a specificity of the primer set may be suspected, and the test may be unsuccessful. You should discuss this problem too.

Author Response

Review #1

Comments and Suggestions for Authors

This review demonstrates that several studies showing that Streptococcus thermophilus can reach the human gastrointestinal tract alive are misinterpreted, because there is a misidentification between S. thermophilus and S. salivarius, an intestinal bacteria, due to no suitable PCR primer set that can be correctly discriminate between both species in feces. It is considered to be socially significant to publish this review, because manufacturers of probiotics need to disclose the correct information to consumers. However, it need some additional discussion.

Major comments

In Line 228-231, a culture independent qPCR method is described. This method should not be used for fecal recovery studies of ingested probiotics, because it detects not only viable cells but also cells that have DNA but cannot form colonies, although there is a problem with the strain specificity of the PCR primer set. A propidium monoazide-qPCR method, which can detect viable cells only, should be used to solve the problem. You should discuss this problem.

Author response: The authors agree to the reviewer comment. Our previous response did not take into consideration the experimental design used in the cited literature (n. 24 in the reference list). We have therefore added a comment in the revised version of the manuscript (lines 231-235);

Not only the problem of S. thermophilus, but the lack of guidelines that demonstrates an appropriate test protocol for the recovery of orally ingested probiotics from feces is considered to be the most important problem. The test protocol should include a double-blind, placebo-controlled crossover design, an appropriate washout period, a species-specific plate culture method, and a strain-specific PCR primer set. And if a probiotics strain is detected in feces of some subjects during the placebo intake period, these subjects should be excluded because of an insufficient washout period or possible compliance violations such as probiotic intake. If a probiotic strain is detected in feces of many subjects during the placebo intake period, a specificity of the primer set may be suspected, and the test may be unsuccessful. You should discuss this problem too.

Author response: we agree with the reviewer observations. We add a comment in the conclusions section (lines 322-330).

Reviewer 3 Report

Line 84: please revise the title for better English, e.g. Existing knowledge on .. survival in the GIT transit conditions.

Line 296: rather than using 'be convinced' please rephrase into e.g. How can we provide adequate scientific evidence towards...

Line 273: please rephrase e.g. Culture-independent detection and identification of... instead of using the word metagenomics that is not always the case bcs many studies are just 16S genomics. 

Author Response

Review #3

Comments and Suggestions for Authors

Line 84: please revise the title for better English, e.g. Existing knowledge on .. survival in the GIT transit conditions.

Author response: We have modified the title as requested.

Line 296: rather than using 'be convinced' please rephrase into e.g. How can we provide adequate scientific evidence towards...

Author response: We have modified the sentence accordingly to the reviewer suggestion.

Line 273: please rephrase e.g. Culture-independent detection and identification of... instead of using the word metagenomics that is not always the case bcs many studies are just 16S genomics.

Author response: We agree with the reviewer comment and we have modified the title accordingly.